# Dynamic and Renormalization-Group Extensions of the Landau Theory of Critical Phenomena

**DOI:** 10.3390/e22090978

**Published:** 2020-09-02

**Authors:** Miroslav Grmela, Václav Klika, Michal Pavelka

**Affiliations:** 1École Polytechnique de Montréal, C.P.6079 suc. Centre-Ville, Montréal, QC H3C 3A7, Canada; 2Department of Mathematics—FNSPE, Czech Technical University in Prague, Trojanova 13, 12000 Prague, Czech Republic; vaclav.klika@fjfi.cvut.cz; 3Mathematical Institute, Faculty of Mathematics, Charles University, Sokolovská 83, 18675 Prague, Czech Republic; pavelka@karlin.mff.cuni.cz

**Keywords:** entropy, critical phenomena, renormalization, multiscale thermodynamics, GENERIC

## Abstract

We place the Landau theory of critical phenomena into the larger context of multiscale thermodynamics. The thermodynamic potentials, with which the Landau theory begins, arise as Lyapunov like functions in the investigation of the relations among different levels of description. By seeing the renormalization-group approach to critical phenomena as inseparability of levels in the critical point, we can adopt the renormalization-group viewpoint into the Landau theory and by doing it bring its predictions closer to results of experimental observations.

## 1. Introduction

Our point of departure is the Landau theory of critical phenomena [1]. We formulate it in two steps highlighting its relation and role in multiscale thermodynamics. The first step is the *2-level formulation of equilibrium thermodynamics*. The first level is the *equilibrium level* with the number of moles *N* and the energy *E* (both per unit volume), serving as state variables. The second level is an *upper level* with the variable *x* serving as the state variable. For example, *x* could be temperature and chemical potential together with an order parameter (as it is in the original formulation of the Landau theory) or it could also be one particle distribution function or other state variables used in mesoscopic theories of macroscopic systems. Equilibrium thermodynamics enters the 2-level formulation in the upper reducing thermodynamic relation (consisting of three real valued functions $S^{↑}\left(x\right),N^{↑}\left(x\right),E^{↑}\left(x\right)$) and in the maximum entropy principle (MaxEnt principle) transforming it (details are in Section 2) to the equilibrium reduced thermodynamic relation S=S(E,N),E=E,N=N. Very often the mesoscopic state variables *x* are fields that enter the specification of $S^{↑}\left(x\right),N^{↑}\left(x\right),E^{↑}\left(x\right)$ as mean fields. The 2-level formulation is therefore often called a *mean-field approach to thermodynamics*.

The second step in the Landau theory is the specification of the upper reducing thermodynamic relation (i.e., specification of the three potentials $S^{↑}\left(x\right),N^{↑}\left(x\right),E^{↑}\left(x\right)$). The particularity of the physics of the macroscopic system under investigation is expressed in these three potentials. Specifications of $S^{↑}\left(x\right),N^{↑}\left(x\right),E^{↑}\left(x\right)$ thus requires commitment to a specific system. For example, in the Gibbs equilibrium statistical mechanics the upper level is the microscopic level with *x* being the n-particle distribution function ($n∼10^{23}$), $S^{↑}\left(x\right)$ is the universal Gibbs entropy, the energy $E^{↑}\left(x\right)$ is the average microscopic energy (expressing the particularity of the system under investigation), and $N^{↑}\left(x\right)$ is the universal potential expressing the number of moles; see, for example [2].

Landau has noted that in the critical region all three potentials $S^{↑}\left(x\right)$, $N^{↑}\left(x\right)$, $E^{↑}\left(x\right)$ tend to be universal. The criticality overrides the particularity of the physical nature of the systems under investigation. Having the universal potentials $S^{↑}\left(x\right),N^{↑}\left(x\right),E^{↑}\left(x\right)$, MaxEnt principle transforms them into a universal equilibrium critical behavior.

The universality of the upper reducing thermodynamic relation in the critical region is based on two observations, one is physical, the other is mathematical. The observation of the physical nature addresses the appearance of criticality on the equilibrium and on the upper levels. While the criticality is more visible, both in experimental observations and in its mathematical representation, on the equilibrium level, it manifests itself also on upper levels. For example, observations of fluctuations in the results of lower-level experimental observations are observations reaching beyond the lower level towards upper levels. Fluctuations appear to be indeed more pronounced in the critical region. The observation of the mathematical nature addresses the universality of potential functions in the critical region investigated in the catastrophe theory [3].

The above two-step formulation of the Landau theory extends naturally to dynamic critical phenomena. In the first step we replace the 2-level formulation of thermodynamics with the 2-level formulation of rate-thermodynamics. The equilibrium level is replaced by a mesoscopic level that still takes into account fewer details than the upper level but it is a level on which the time evolution takes place. We recall that no time evolution takes place on the equilibrium level. In fact, we choose to use the vector field governing the lower time evolution as the state variable on the lower level. The rate-thermodynamics on the upper level is expressed in the upper reducing rate-thermodynamic relation $\left(\Sigma^{↑}\left(x\right),Y^{↑}\left(x\right)\right)$, where $\Sigma^{↑}\left(x\right)$ is the rate entropy, Y(x) is the lower level vector field expressed in terms of *x*. The MaxRent principle (Maximum Rate Entropy principle—see details in Section 3), replacing the MaxEnt principle in equilibrium thermodynamics, transforms then the upper reducing rate-thermodynamic relation to the lower reduced rate-thermodynamic relation (Σ(Y),Y), where Σ(Y) is the lower rate entropy and *Y* is the lower vector field.

Regarding the comparison of predictions of Landau’s theory with results of experimental observations, the agreement is only qualitative. The problem is in the multiscale nature of critical phenomena. The closer is the critical point, the closer is the upper level to the lower level and in the critical point itself all levels become inseparable. This observation is then taken as a basis for the renormalization-group theory of critical phenomena [4]. In Section 4 we extend the Landau theory to 3-level formulation, which then provides a setting for the renormalization-group theory of critical phenomena seen as an extension of the Landau theory.

## 2. Landau’s Theory of Static Critical Phenomena

*Level of description* is an autonomous collection of results of certain type of experimental observations (different for different levels) together with a model that allows to organize them, to reproduce them, and to make predictions. The model, based on the insight inspired by the experimental data and by investigating relations to nearby levels involving less or more details, offers also an understanding of the physics involved. For instance, the *equilibrium level* with the energy *E*, number of moles *N*, and volume *V* serving as state variables and the *microscopic level* with position and momenta of ∼10${}^{23}$ particles composing the macroscopic system serving as state variable are examples of two different autonomous levels of description. The latter is more microscopic (it takes into account more details) than the former. We call the latter level an *upper level* and the former the *lower level*. In this section the lower level will always be the equilibrium level. The state variable on the upper level is denoted by the symbol *x*. For example, in the Landau theory *x* is usually the equilibrium temperature and an appropriately chosen order parameter. On the level of kinetic theory x=f(r,v) or on the level of hydrodynamics x=(ρ(r),e(r),u(r)), where f(r,v) is one particle distribution function, ***r*** is the position vector and ***v*** momentum of one particle; ρ(r) is the field of mass density, e(r) the field of internal energy, u(r) the field of momentum. The state space on the equilibrium level is denoted by the symbol M, i.e., (E,N,V)∈M, the state space on the upper level is denoted by the symbol $M^{↑}$, i.e., $x\in M^{↑}$.

Every level is autonomous and differs from other levels in the amount of details (that are taken into account in both experimental observations and the mathematical formulation) and in the range of applicability. However self-contained are the levels, their mathematical formulation is closely related to their relationship to other levels. From investigating relations to upper levels (i.e., to levels involving more details) comes a structure that we shall call *reduced structure* and from investigating relations to lower levels (i.e., to levels involving fewer details) comes the *reducing structure*. Both structures equip the state space with a geometry and a vector field. The geometry is a mathematical formulation of thermodynamics. Every level has thus reduced and reducing thermodynamics and reduced and reducing vector fields. Below, we limit ourselves only to the reducing thermodynamics on the upper level and the reduced thermodynamics on the lower level (which is in this section the equilibrium level).

We emphasize that the term “reduction” has in this paper the same meaning as “emergence”. Some details on the upper level are lost in the reduction from an upper level to a lower level but at the same time an emerging overall pattern is gained. The process of reduction, as well as processes conductive to an emergence of overall features (pattern-recognition processes), involve both a loss and a gain. The terms “upper” and “lower” levels that we use in this paper have a different meaning than they have in, say, social sciences. The lower level is inferior from the upper level in the amount of details but superior in the ability to see overall patterns.

### 2.1. 2-Level Equilibrium Thermodynamics

Among many questions about the origin of both reduced and reducing structures and about their relations, we shall discuss only the one that is directly relevant to the Landau theory. We shall investigate the passage from the reducing thermodynamics on the upper level to the reduced thermodynamics on the equilibrium level. A few comments about the placement of the investigation of this passage in the larger context of multiscale thermodynamics are discussed at the end of this section.

The upper reducing thermodynamic relation
(1)$$S^{↑}\left(x\right),E^{↑}\left(x\right),N^{↑}\left(x\right)$$
is one of several possible forms of the mathematical formulation of the upper reducing thermodynamics. The quantities introduced in (Equation 1) are the upper energy per unit volume $E^{↑}:M^{↑}\to R$, the upper number of moles per unit volume $N^{↑}:M^{↑}\to R$, and the upper reducing entropy per unit volume $S^{↑}:M^{↑}\to R$ and are assumed to be sufficiently regular.

The reduced equilibrium thermodynamic relation
(2)S(E,N),E,N
is obtained from (Equation 1) by the following reducing Legendre transformation. We make this transformation in four steps.

Step 1: We introduce upper reducing thermodynamic potential (Note that *x* can be any state variable, e.g., distribution function, hydrodynamic fields, electromagnetic fields, see [5])
(3)$$\Phi^{↑}\left(x;E^{*},N^{*}\right)=-S^{↑}\left(x\right)+E^{*}E^{↑}\left(x\right)+N^{*}N^{↑}\left(x\right)$$
where $\left(E^{*},N^{*}\right)$ are conjugate equilibrium state variables. In the standard equilibrium thermodynamic notation $E^{*}=\frac{1}{T}$, where *T* is the equilibrium temperature, and $N^{*}=-\frac{\mu }{T}$, where μ is the equilibrium chemical potential.

Step 2: We solve the equation
(4)$$\Phi_{x}^{↑}=0$$We use hereafter the notation: $\Phi_{x}^{↑}=\frac{\partial \Phi^{↑}}{\partial x}$, where $\frac{\partial }{\partial x}$ is an appropriate functional derivative if *x* is a function (i.e., an element of an infinite dimensional space). Let $\hat{x}\left(E^{*},N^{*}\right)$ be the solution to (Equation 4).

Step 3: We introduce
(5)$$S^{*}\left(E^{*},N^{*}\right)=\Phi^{↑}\left(\hat{x}\left(E^{*},N^{*}\right);E^{*},N^{*}\right)$$
called a reduced conjugate entropy.

Step 4: Finally, we pass from $S^{*}\left(E^{*},N^{*}\right)$ to S(E,N) by the Legendre transformation (i.e., we introduce the equilibrium thermodynamic potential $\Phi^{*}\left(E^{*},N^{*};E,N\right)=-S^{*}\left(E^{*},N^{*}\right)+EE^{*}+NN^{*}$, solve $\Phi_{E}^{*}=0$, $\Phi_{N}^{*}=0$, and arrive at $S\left(E,N\right)=\Phi^{*}\left({\hat{E}}^{*}\left(E,N\right),{\hat{N}}^{*}\left(E,N\right);E,N\right)$, where $\left({\hat{E}}^{*}\left(E,N\right),{\hat{N}}^{*}\left(E,N\right)\right)$ are solutions to $\Phi_{E}^{*}=0$, $\Phi_{N}^{*}=0$.

The reducing Legendre transformation (Equation 1) → (Equation 2) can also be seen as maximization of the upper reducing entropy $S^{↑}\left(x\right)$ subjected to constraints $E^{↑}\left(x\right),N^{↑}\left(x\right)$ [5,6]. The conjugate equilibrium state variables $E^{*},N^{*}$ play the role of Lagrange multipliers. This viewpoint then gives the passage (Equation 1) → (Equation 2) the name Maximum Entropy principle (MaxEnt principle)

Summing up, the MaxEnt passage from the upper level to the equilibrium level, via the upper reducing thermodynamic relation (Equation 1), is the following sequence of two mappings
(6)$$\left(S^{↑}\left(x\right),E^{↑}\left(x\right),N^{↑}\left(x\right)\right)\mapsto \left(S^{*}\left(E^{*},N^{*}\right),E^{*},N^{*}\right).\mapsto \left(S\left(E,N\right),E,N\right)$$
The second mapping in (Equation 6) is the standard Legendre transformation. The first mapping is the upper reducing Legendre transformation expressing the MaxEnt principle.

In the particular case when x=(E,N) and $N^{↑}\left(E,N\right)=N,E^{↑}\left(E,N\right)=E$, there is no reduction in (Equation 6) and both arrows in (Equation 6) are (one-to-one) standard Legendre transformations:(7)$$\left(S\left(E,N\right),E,N\right)\mapsto \left(S^{*}\left(E^{*},N^{*}\right),E^{*},N^{*}\right)\mapsto \left(S\left(E,N\right),E,N\right)$$

To conclude this section we turn to questions like where the potentials $S^{↑}\left(x\right),E^{↑}\left(x\right),N^{↑}\left(x\right)$ come from and why the upper entropy $S^{↑}\left(x\right)$ is maximized subjected to constraints $E^{↑}\left(x\right)$ and $N^{↑}\left(x\right)$. These questions are answered simply by the existence of the autonomous upper level and the existence of the autonomous equilibrium level. The autonomous existence implies that there exists a way to prepare macroscopic systems for their investigations on the equilibrium level and that the time evolution describing the preparation process can be formulated on the upper level as a reducing time evolution. It is in this reducing time evolution where the potentials $S^{↑}\left(x\right),E^{↑}\left(x\right),N^{↑}\left(x\right)$ make their first appearance. The entropy $S^{↑}\left(x\right)$ generates the reducing time evolution. Maximization of $S^{↑}\left(x\right)$ reflects the property of its solutions expressing mathematically the approach to the equilibrium level. We note that in the context of the classical formulation of equilibrium thermodynamics the existence of the preparation process for the equilibrium level (the existence of equilibrium states) is a subject of the zero axiom (see [2]). The 2-level formulation of equilibrium thermodynamics can be thus seen as a way to bring the zero axiom to an active participation in the formulation of equilibrium thermodynamics.

### 2.2. 2-Level Equilibrium Thermodynamics in the Critical Region

Equilibrium-level experimental observations of phase transitions are mathematically expressed in various types of singularities of the equilibrium reduced thermodynamic relation S(E,N). We call the subspace of the equilibrium level state space at which the Hessian (matrix of second derivatives) of S(E,N) has a nontrivial nullspace as a critical submanifold. Its neighborhood is called a critical region.

Experimental observations made on upper levels (i.e., levels involving more details than the equilibrium level) show that the critical behavior seen on the equilibrium level is also seen on the upper levels. For instance, it is well established that fluctuations in the results of the equilibrium-level measurements (that is an example of measurements that involve more details than the equilibrium-level measurements) become very pronounced in the critical region. The mathematical manifestation of the criticality in the upper reducing thermodynamic relation $S^{↑}\left(x\right),E^{↑}\left(x\right),N^{↑}\left(x\right)$ is however different from its mathematical manifestation on the equilibrium level. The potentials $S^{↑}\left(x\right),E^{↑}\left(x\right),N^{↑}\left(x\right)$ remain completely smooth, but Equation (Equation 4) has two or more solutions.

### 2.3. Van der Waals Theory

We now illustrate the Landau theory on the van der Waals theory of a gas composed of particles interacting via long range attractive and short range repulsive forces. The macroscopic physical system investigated in this illustration is a gas composed of particles interacting via long range attractive forces and short range repulsive forces (van der Waals gas). The latter forces are treated as constraints and their influence enters the entropy rather than energy. The mathematical model of this system on the equilibrium level is the well known classical van der Waals model, on the level of kinetic theory the van Kampen model [7] (see also [8]) and its dynamical extension ([9]).

The upper level is the level of kinetic theory with the one particle distribution function
(8)x=f(r,v)
serving as a single state variable, ***r*** is the position coordinate and ***v*** the momentum of one particle. In this example we specify explicitly the upper reducing thermodynamic relation $S^{↑}\left(x\right),E^{↑}\left(x\right),N^{↑}\left(x\right)$ by using arguments developed mainly in the Gibbs equilibrium statistical mechanics. Having $S^{↑}\left(x\right),E^{↑}\left(x\right),N^{↑}\left(x\right)$, we identify the critical point and subsequently restrict $S^{↑}\left(x\right),E^{↑}\left(x\right),N^{↑}\left(x\right)$ to the critical region. The resulting potentials take the form of the Landau critical thermodynamic potentials. This illustration has already been presented in [8], we can therefore omit details.

Following van Kampen [7], the upper reducing thermodynamic relation (Equation 1) representing on the level of kinetic theory the van der Waals gas is
(9)$$\begin{array}{ccc} E^{↑}\left(f\right) & = & \int dr\int dv\left(\frac{v^{2}}{2}f\left(r,v\right)+\frac{1}{2}\int dr_{1}\int dv_{1}V_{pot}(|r-r_{1}\left|\right)f\left(r,v\right)f\left(r_{1},v_{1}\right)\right)\\ N^{↑}\left(f\right) & = & \int dr\int dvf(r,v)\\ S^{↑}\left(f\right) & = & \int dr\int dv\left(-f\left(r,v\right)\ln f\left(r,v\right)-f\left(r,v\right)\frac{\partial \theta }{\partial n(r)}\right) \end{array}$$
where we put the volume of the region in which the van der Waals gas is confined equal to one, the mass of one particle is also equal to one, $V_{pot}(|r-r_{1}\left|\right)$ is the potential energy,
(10)$$\begin{array}{ccc} n(r) & = & \int dvf(r,v)\\ \theta (n(r)) & = & \frac{1-Bn(r)}{B}\left(\ln\left(1-Bn\left(r\right)\right)-1\right) \end{array}$$
and where B∈R is a small (proportional to the volume of one particle) parameter. Note that n(r)=∫dvf(r,v) is the local particle density. In $E^{↑}$, the first term is the kinetic energy, the second the potential energy. In $S^{↑}$, the first term is the Boltzmann entropy, the second term is the contribution to the entropy due to the excluded volume constraint.

By making the transformations in (Equation 6), we arrive at the classical well known van der Waals thermodynamic relation (see details in [5] (pp. 43–44) and [8]).

Now we proceed to investigate the critical region. First note that van Kampen [7] showed that the critical points corresponding to the end points of the critical curve in van der Waals gas are spatially homogeneous. This knowledge can be translated within this multiscale framework formulation into a restriction of the MaxEnt distribution function and the reduction to the level with state variable *n* (see the Appendix A and [8] for details). In short, we begin by looking for solutions of (Equation 4) only among n(r) that are independent of ***r***. With this restriction, we arrive at
(11)$$\Phi^{↑}\left(n;\alpha ,\beta \right)=n\ln n+n\frac{d\theta }{dn}-\frac{1}{2}\beta V_{pot}n^{2}-\left(\alpha -\frac{3}{2}\ln\frac{\beta }{2\pi }\right)n$$
where we use the shorthand notation $\beta =\frac{1}{T}$, $\alpha =-\frac{\mu }{T}$ and $V_{pot}=\int dr_{1}V_{pot}(|r-r_{1}\left|\right)$.

The critical point is

A MaxEnt value: an extremum of reducing thermodynamic potential that governs the evolution and hence this extremum corresponds to an equilibrium state;A point where loss of convexity occurs: the extremum (equilibrium point) is ambiguous, multiple or continuum of extrema are plausible;Critical point of the whole system (including the inverse temperature $\beta =\frac{1}{T}$), i.e., the lowest temperature (*T*) and chemical potential (α) for which a critical point given by the above two points still exists: an extremal point, i.e., the third derivative with respect to *n* has to vanish as follows from Taylor expansion and the fact that the two above requirements can be translated into vanishing first two derivatives.

In short, the critical point is a point $\left(n_{c},\alpha_{c},\beta_{c}\right)$ where the potential has a stationary point, just loses convexity and has a minimum. By taking the Taylor expansion about the critical point,
(12)$$\begin{array}{cc} \Phi^{↑}\left(n,\alpha_{c},\beta_{c}\right) & =\Phi^{↑}\left(n_{c},\alpha_{c},\beta_{c}\right)+\frac{\partial \Phi^{↑}}{\partial n}\delta n+\frac{1}{2!}\frac{\partial^{2}\Phi^{↑}}{\partial n^{2}}{\left(\delta n\right)}^{2}\\ \; & \;+\frac{1}{3!}\frac{\partial^{3}\Phi^{↑}}{\partial n^{3}}{\left(\delta n\right)}^{3}+\frac{1}{4!}\frac{\partial^{4}\Phi^{↑}}{\partial n^{4}}{\left(\delta n\right)}^{4}+O{\left(\delta n\right)}^{5}, \end{array}$$
the requirements on the critical point lead to equations
(13)$$\Phi_{n}^{↑}=0;\Phi_{nn}^{↑}=0;\Phi_{nnn}^{↑}=0;\Phi_{nnnn}^{↑}>0.$$

Hence the critical point of van der Waals gas is given by
(14)$$\begin{array}{ccc} & & n_{c}=\frac{1}{3B};\;\;\;\beta_{c}=\frac{27B}{4V_{pot}}\\ & & \alpha_{c}=\frac{1}{2}\ln\left(3B\right)+\frac{3}{2}\ln\frac{B}{V_{pot}}+\frac{3}{4}+4\ln\frac{3}{2}-\frac{3}{2}\ln\left(2\pi \right). \end{array}$$

See Appendix A for more details.

We now make an explicit choice of the order parameter
$$\xi =n-n_{c},$$
where $n_{c}$ is the critical value of *n*, and define
(15)$$\Phi_{crit}^{↑}\left(n;\omega_{1},\omega_{2},\omega_{3}\right)=\Phi \left(n;\alpha ,\beta \right)-\Phi \left(n_{c};\alpha ,\beta \right).$$

With the explicit knowledge of the reducing fundamental thermodynamic potential (Equation 11) we know that in a neighborhood of the critical point we arrive at
(16)$$\Phi_{crit}^{↑}\left(\xi ;\omega_{1},\omega_{2},\omega_{3}\right)=\omega_{1}\xi +\frac{1}{2}\omega_{2}\xi^{2}+\frac{1}{24}\omega_{3}\xi^{4},$$
where
(17)$$\begin{array}{ccc} & & \omega_{1}=a_{1}\left(\alpha -\alpha_{c}\right)+a_{2}\left(\beta -\beta_{c}\right)\\ & & \omega_{2}=a_{3}\left(\beta -\beta_{c}\right) \end{array}$$
as the coefficient of α in $\Phi^{↑}$ is linear in *n* while the coefficient of β is quadratic in *n*. Note that the cubic term is missing in the expansion as its coefficient is independent of α,β and hence $\Phi_{nnn}^{↑}\left(n_{c};\alpha ,\beta \right)=\Phi_{nnn}^{↑}\left(n_{c};\alpha_{c},\beta_{c}\right)=0$ due to (Equation 13). This form of the thermodynamic relation in the critical region $\Phi_{crit}^{↑}$ can be put in a more general framework due to Landau [10] (Chapter XIV). Expressions for $a_{1},a_{2},a_{3},\omega_{3}$ involve the parameters $B,V_{pot}$, serving as the material parameters in the van der Waals theory.

From (Equation 16) we obtain
(18)$$S_{crit}^{↓*}\left(\alpha ,\beta \right)=\Phi_{crit}^{↑}\left(\hat{\xi }\left(\alpha ,\beta \right);\alpha ,\beta \right)$$
where $\hat{\xi }\left(\alpha ,\beta \right)$ is a solution to
(19)$${\left(\Phi_{crit}^{↑}\right)}_{\xi }=0.$$

The lower reduced entropy (Equation 18) provides complete information about the behavior (the behavior seen in equilibrium-thermodynamic observations) of the van der Waals gas in a small neighborhood of the critical point. In particular, we obtain the critical exponents arising in the dependence of $S_{crit}^{↓*}\left(\alpha ,\beta \right)$ on α and β.

A simple way to see that $S_{crit}^{↓*}\left(\alpha ,\beta \right)$ is a generalized homogeneous function and thus to identify the critical exponents is to use (Equation 16) with $\omega_{1,2,3}$ now being the variables instead of α,β. We replace ξ in (Equation 16) with $\lambda^{-1/4}\xi$. We obtain
$$\begin{array}{ccc} & & \Phi_{crit}^{↑}\left(\lambda^{-1/4}\xi ;\omega_{1},\omega_{2},\omega_{3}\right)=\lambda^{-1}\left(\left(\omega_{1}\lambda^{3/4}\right)\xi +\left(\omega_{2}\lambda^{1/2}\right)\xi^{2}+\omega_{3}\xi \right)\\ & & =\lambda^{-1}\Phi_{crit}^{↑}\left(\xi ;\left(\omega_{1}\lambda^{3/4}\right),\left(\omega_{2}\lambda^{1/2}\right),\omega_{3}\right) \end{array}$$
and consequently, noting $\omega_{3}$ is unaffected by the rescaling,
(20)$$S_{crit}^{↓*}\left(\omega_{1},\omega_{2}\right)=\lambda^{-1}S_{crit}^{↓*}\left(\lambda^{3/4}\omega_{1},\lambda^{1/2}\omega_{2}\right).$$

Finally, one could invert $\left(\omega_{1},\omega_{2}\right)$ from (Equation 17) to get a (generalized) scaling for $S_{crit}^{↓*}\left(\alpha ,\beta \right)$.

Still another view of this relation can serve as an introduction to the renormalization-group theory of critical phenomena discussed below in Section 4. We start again with (Equation 16) and write it in the form $\Phi_{crit}^{↑}\left(\xi ;\omega \right)$, where $\omega =(\omega_{1},\omega_{2},\omega_{3})$ is given in (Equation 17). Our aim is to introduce a *renormalization time evolution* (i.e., renormalization group of transformations generated by a vector field) of ω and of $\Phi_{crit}^{↑}$ such that: $$\Phi_{crit}^{↑}\left(\xi ;\omega \left(\tau \right),\tau \right)=\Phi_{crit}^{↑}\left(\xi ;\omega \right)\;\;\forall \;\;\tau >0$$
with the initial conditions
(21)$$\begin{array}{ccc} \omega (0) & = & \omega \\ \Phi_{crit}^{↑}\left(\xi ;\omega ,0\right) & = & \Phi_{crit}^{↑}\left(\xi ;\omega \right) \end{array}$$
and the constraint
(22)$$\omega_{3}\left(\tau \right)=\omega_{3}\;\;\;\forall \;\;\tau >0$$

The renormalization time is denoted by the symbol τ. We emphasize that the renormalization time τ has nothing to do with the real time *t*. The renormalization time evolution will become the basis for a new definition of critical points discussed in Section 4. From the physical point of view, the constraint expresses the requirement that the material parameter *B* entering the repulsive short range forces in (Equation 10) remains unchanged in the renormalization process.

We begin with
(23)$${\left(\Phi_{crit}^{↑}\right)}_{\tau }=-\chi \Phi_{crit}^{↑}$$
with χ>0 being at this point an unspecified parameter and with the initial condition given by the second line in (Equation 21). It can be easily verified that [8]
(24)$$\dot{\omega }=R\left(\chi ,\omega \right)$$
with the initial condition given by the first line in (Equation 21) and
(25)$$R\left(\chi ,\omega \right)=\left(\begin{array}{ccc} \chi -1 & 0 & 0\\ 0 & \chi -2 & 0\\ 0 & 0 & \chi -4 \end{array}\right)\omega^{T}.$$

We see now that with χ=4 we satisfy both (Equation 21) and the constraint (Equation 23).

*The fixed point of the renormalization time evolution is the critical point and the eigenvalues of the vector field linearized about the fixed point are the critical exponents*.

This statement, which has arisen as a simple observation in the particular context discussed above, is in fact a definition of the critical points and the critical exponents in the renormalization-group theory of critical phenomena (see Section 4). In the case of (Equation 25) the linearization is, of course, unnecessary since the vector field is already linear.

Finally we compare the classical analysis of the van der Waals gas with the analysis based on the Landau theory. The starting point of the classical analysis is the physical insight that led us to the upper reducing thermodynamic relation (Equation 9). By restricting it to the critical region we have arrived at the Landau expression (Equation 16). The starting point of the Landau theory is the expression (Equation 16). The quantity ξ, called in the Landau theory an order parameter, does not need to have a specific physical interpretation, nor the coefficients $a_{1},a_{2},a_{3}$ are specified in the Landau theory.

The extra information about the critical phenomena that the classical van der Waals theory provides (but only for the van der Waals gas) is thus: (i) the location of the critical point in the state space $M^{↑}$, (ii) physical interpretation of the order parameter, (iii) a detailed knowledge of the critical behavior beyond a small neighborhood of the critical point. On the other hand, the advantage of the Landau theory is its universal applicability. In Section 4 we make a comment about the renormalization group theory, the objective of which is to bring the critical exponents implied by the van der Waals theory (and thus also the Landau theory) closer to those seen in experiments.

Before leaving the van der Waals theory, we mention that the static version of the theory recalled above has been upgraded to the dynamical theory in [9]. The kinetic equation of which solutions make the maximization of the entropy $S^{↑}\left(f\right)$ subjected to constraints $E^{↑}\left(f\right),N^{↑}\left(f\right)$ (see (Equation 9)) is the Enskog Vlasov kinetic equation.

## 3. Landau’s Theory of Dynamic Critical Phenomena

In the 2-level formulation of the equilibrium thermodynamics we replace the equilibrium level with a lower level that still takes into account fewer details than the upper level, but it is a mesoscopic level on which the time evolution, called a lower time evolution, takes place. We recall that no time evolution takes place on the equilibrium level that served us as the lower level in the preceding section. We again assume that both the upper and the lower levels are well established (well tested with experimental observations) autonomous levels. This then means that by investigating solutions to the upper time evolution equations we have to be able to split the upper time evolution into a reducing time evolution describing the preparation process for using the lower level and a reduced time evolution that is the lower time evolution. The investigation leading to the split is essentially a pattern recognition process in solutions to the upper governing equations.

There are two types of the reducing and the reduced time evolutions. The reducing time evolution can be either the time evolution taking place in $M^{↑}$ and approaching an invariant (or in most cases a quasi-invariant) manifold $M^{↓}\subset M^{↑}$ that represents in $M^{↑}$ the state space $M^{↓}$ used on the lower level or it can be the time evolution of vector fields $Y^{↑}\left(x\right)\in X\left(M^{↑}\right)$ taking the vector field generating the upper time evolution to the vector field generating the lower time evolution. The former viewpoint is discussed for example in [11,12,13,14]. In this paper we follow the second route, discussed in [15], since on this route we can directly transpose the 2-level equilibrium thermodynamics introduced in the previous section to 2-level rate-thermodynamics. We use "rate" to point out that the state space is the space of vector fields.

The upper reducing rate-thermodynamic relation
(26)$$\Sigma^{↑}\left(x\right),Y^{↑}\left(x\right)$$
replaces the upper reducing thermodynamic relation (Equation 1). The passage from the upper reducing rate-thermodynamic relation (Equation 26) to the lower reduced rate-thermodynamic relation
(27)Σ(Y),Y
remains the same as the passage from the upper reducing thermodynamic relation (Equation 1) to the lower reduced thermodynamic relation (Equation 2) in 2-level thermodynamics, discussed in Section 2. We introduce an upper reducing rate-thermodynamic potential
(28)$$\Psi^{↑}\left(x;Y^{*}\right)=-\Sigma^{↑}\left(x\right)+\langle Y^{*},Y^{↑}\left(x\right)\rangle$$
where $Y^{*}$ are conjugate lower vector fields. The sequence of mappings
(29)$$\left(\Sigma^{↑}\left(x\right),Y^{↑}\left(x\right)\right)\mapsto \left(\Sigma^{↓*}\left(Y^{*}\left(y\right)\right),Y^{*}\left(y\right)\right)\mapsto \left(\Sigma^{↓}\left(Y\left(y\right)\right),Y\left(y\right)\right)$$
corresponds in MaxRent to the sequence of mappings (Equation 6) in MaxEnt. The lower vector field is $Y\left(y\right)=\Sigma_{Y^{*}\left(y\right)}^{↓*}$.

How do we specify the upper reducing relations (Equation 1) or (Equation 26)? The following three routes can be taken.

(i) Both relations (Equation 1) and (Equation 26) arise from a detail investigation of the upper time evolution. Since both the upper and the lower levels are well established, the upper level has to reduce, by following a certain preparation process in which time evolution is described by the reducing time evolution, to the lower level. The reducing time evolution then introduces the upper reducing thermodynamics relation (Equation 1) or upper reducing rate-thermodynamic relation (Equation 26). In this paper we do not introduce and discuss explicitly the reducing time evolution, neither in the equilibrium thermodynamics nor in the rate-thermodynamics. It is important to recall that the upper reducing thermodynamic relation (Equation 1) representing an ideal gas on the level of kinetic theory has been originally obtained by Boltzmann from analyzing solutions (Boltzmann’s H-theorem) of the Boltzmann kinetic equation describing the reducing time evolution. In the Boltzmann analysis the kinetic equation is primary, and the Boltzmann entropy arises as a result.

(ii) In the critical region the upper reducing thermodynamic potentials (Equation 3) and (Equation 28) are determined by mathematical results arising in the catastrophe theory [3].

(iii) The association between specific physical systems and the upper reducing thermodynamic relations (Equation 1) in equilibrium thermodynamics can also be investigated by physical arguments developed mainly in the Gibbs equilibrium statistical mechanics (as we did in the illustration in Section 2.3).

Before proceeding to the illustration we make a few remarks.

Our investigation in the preceding section was limited to equilibrium. We have considered only systems that are allowed to reach equilibrium states. Behavior of macroscopic systems that are prevented from reaching the equilibrium states (either by external or internal forces) cannot be described on the equilibrium level but can be described on a lower level. For instance the experimentally observed behavior of a Rayleigh–Bénard system (a thin horizontal layer of a fluid heated from below) is well described on the level of hydrodynamics (in Boussinesq equations) [16,17]. The lower reduced thermodynamic relation that we are getting on the lower level from relating it to an upper level provides thus thermodynamics also for such externally or internally forced systems [18].

The equilibrium can be reached either directly (*upper level* → *equilibrium level*) or indirectly (*upper level* → *lower level* → *equilibrium level*). We require that the equilibrium thermodynamic relations obtained by following both routes are identical for consistency in the multilevel framework. This requirement implies the following relation between quantities entering the equilibrium and rate-thermodynamic relations
(30)$${\dot{S}}^{↑}\left(y\right)={\left[\langle Y^{*},\Sigma_{Y^{*}}^{↓*}\rangle \right]}_{Y^{*}=S_{y}^{↑}\left(y\right)}$$

By *y* we denote the state variables on the lower level, $S^{↑}\left(y\right)$ is the upper entropy generating the approach from the lower level to the equilibrium level. The relation (Equation 30) makes precise the connection between the rate entropy $\Sigma^{↑}\left(x\right)$ on the upper level and the entropy production ${\dot{S}}^{↑}\left(y\right)$ on the lower level.

### 3.1. Illustration: Immiscible Fluids

Dispersions of two immiscible fluids (fluid A and fluid B) have two essentially different morphologies. One in which the fluid A is dispersed in the form of droplets in the fluid B that forms a continuous phase. The second is the inverse, fluid B is dispersed and fluid A is continuous. The transition between these two morphologies is a dynamic critical point called phase inversion. Dispersions under consideration are subjected to externally imposed flows.

As it was in the case of the van der Waals gas in Section 2.3, we want to identify the critical point (in particular the critical concentrations) and to investigate the behavior in the critical region (in particular the flow behavior of the dispersion). As we saw in the case of the investigation of static critical phenomena in Section 2.3, both questions are answered if we know explicitly the upper reducing thermodynamic potential. In the case of phase inversion it would be the explicit knowledge of the upper reducing rate-thermodynamic potential (Equation 28). In general, the problem of finding thermodynamic potentials corresponding to specific physical systems is more difficult in rate-thermodynamics than in thermodynamics. For example in the specification of the van der Waals upper reducing thermodynamic potential (Equation 11) we have used the insight offered by Gibbs investigations in which the upper level is the Microscopic level and the lower level the equilibrium level. No such powerful source of insights seems to be available in rate-thermodynamics. Nevertheless, we know that the rate-thermodynamic potentials exist. This is because the upper and the lower levels exist as autonomous levels and consequently the upper level approaches the lower level. In the case of dispersions the lower level is the level of hydrodynamics and the upper level can be, for instance, the Microscopic level or it could also be the level of kinetic theory. Just the knowledge of the existence of the upper and lower levels gives us the right to use the Landau theory (which will address the behavior in the critical region) and also certain arguments that are based on partial knowledge rate-thermodynamic potentials that will address the problem of identifying the point of phase inversion.

We turn first to the latter investigation. In the absence of a complete knowledge of the rate-thermodynamic potentials, we can attempt to identify them separately on both sides of the phase inversion. At the point of phase inversion the two potentials must be equal. Their equality is then an equation determining the point of phase inversion. With the surface energy playing the role of the potentials, this analysis has been made in [19,20] and with the rate-thermodynamic potential (Equation 28) in [21]. We have seen in (Equation 30) that the upper reducing rate-thermodynamic potential is related to but not identical to the entropy production. In [21] the entropy production on both sides of the phase inversion is specified and then put (as an approximation) on the place of rate-thermodynamic potentials.

The Landau theory has been applied to the problem of phase inversion in [22]. The order parameter is an unspecified characterization of the morphology of the dispersion. It can be for instance an average (oriented) curvature of the interface separating the two fluids.

### 3.2. Illustration: Shear Banding

It was experimentally observed in [23] that the Taylor–Couette flow of a special shear banding fluid exhibits unusual behavior. In the experiments either the force required to rotate the outer cylinder (shear stress) or speed of the rotation (shear rate) can be controlled, the other being measured. It turns out that when varying shear stress, shear rate behaves continuously while when varying shear rate, shear stress exhibits a jump. Such behavior, also called dissipative phase transition, can be captured by a non-convex dissipation potential giving relation between shear stress and shear rate [24].

In rate thermodynamics (also called CR-thermodynamics [21]) the roles of state and conjugate variables are played by thermodynamic forces and fluxes (or vice versa), and the role of entropy is played by dissipation potential. In order to see a phase transition in the rate thermodynamics (as in the above-introduced experiment), one thus needs to be equipped with a non-convex dissipation potential. Such potential was proposed in [24] for a dissipative phase transition in complex fluids,
(31)$$\Xi =0.01x^{2}+\frac{1}{2}-\frac{1}{2(1+x^{2})}.$$

Note that the potential is written in a non-dimensional form and that *x* represents the norm of the deviatoric stress tensor. The dissipation potential is clearly non-convex, as is apparent from Figure 1.

In order to obtain the stress tensor, one has to perform the Legendre transformation
(32)$$\frac{\partial \Phi }{\partial x}=0\;\mathrm{for}\;\Phi \left(x,\gamma \right)=-\Xi \left(x\right)+\gamma x,$$
where γ represents the shear rate.

Let us now find the critical points (there are two) of potential Φ. The loss of convexity at the critical points is expressed by the equation
(33)$$\Phi_{xx}=0,$$
the solutions of which are the critical stresses $x_{c1}=3.24294$ and $x_{c2}=0.591376$. The critical shear rates are then obtained by solving the equations
(34)$$\Phi_{x}\left(x_{c1},\gamma_{c1}\right)=0\;\mathrm{and}\;\Phi_{x}\left(x_{c2},\gamma_{c2}\right)=0,$$
giving $\gamma_{c1}=0.0893092$ and $\gamma_{c2}=0.336446$. The first and second derivatives at the critical point vanish. Note, however, that the third derivative does not vanish and the critical points are thus not minima of the potential Ψ and the rate thermodynamics is thus not stable in the critical points.

We can further expand the potential Ψ in power series in the critical points,
(35)$$\begin{array}{cc} \Psi (\gamma_{c1},\xi ) & \approx -0.272127-0.0035087\xi^{3}+0.0011077\xi^{4}\;\mathrm{for}\;\xi =x-x_{c1} \end{array}$$
(36)$$\begin{array}{cc} \Psi (\gamma_{c2},\xi ) & \approx 0.0659143+0.231744\xi^{3}-0.210484\xi^{4}\;\mathrm{for}\;\xi =x-x_{c2}. \end{array}$$

Shifting the potential around the critical point by a constant value so that their value at the critical point is zero, the general expansion around the critical point (also in the direction of the parameter γ) then reads
(37)$$\begin{array}{cc} \Psi (\gamma_{c1},\xi ) & \approx \omega_{1}\left(\gamma -\gamma_{c1}\right)\xi^{2}\;\mathrm{for}\;\xi =x-x_{c1} \end{array}$$
(38)$$\begin{array}{cc} \Psi (\gamma_{c2},\xi ) & \approx \omega_{2}\left(\gamma -\gamma_{c2}\right)\xi^{2}\;\mathrm{for}\;\xi =x-x_{c2}. \end{array}$$The first derivative disappears, since we have one parameter γ that can be used to keep it zero, but the coefficients in front of the second derivatives only disappear in the critical points.

In summary, the geometric analysis of critical phenomena can be carried out also in the realm of rate thermodynamics, where the thermodynamic forces and fluxes play the role of state and conjugate variables. The universal behavior near the critical points is observed similarly as in the classical theory.

## 4. Renormalization-Group Theory of Critical Phenomena in the Setting of Landau’s Theory

The universality of the upper reducing thermodynamic relations in the critical region is based on mathematical arguments [3]. The mathematical universality then implies the universality of physical behavior that can be observed experimentally. Is the experimentally observed critical behavior indeed universal? The answer is well known. Predictions of the Landau theory agree with results of experimental observations only qualitatively. How can we explain it?

The problem is in the autonomy of levels in the critical region. The closer the critical point, the more difficult it is to separate the levels. This general observation is often illustrated on the example of the observation of fluctuations. We recall that fluctuations seen in results of experimental observations made on level L are in fact observations that reach beyond the L-level towards observations belonging to a level involving more details. This means that large fluctuations seen on the level L indicate that the level L ceases to be autonomous. Some of the details ignored on the level L cannot be ignored anymore in order to keep the level L autonomous. In the critical point itself the levels become inseparable. This feature of criticality is then taken as the basis for the renormalization-group theory of critical phenomena.

Our objective in this section is to formulate the renormalization-group theory of critical phenomena as an extension of the Landau theory. For the sake of simplicity we make below the extension only for the Landau theory of static critical phenomena. Its dynamical version will be the subject of a future paper.

The first step in the extension is a replacement of the upper level with more upper levels. In the enlarged family of upper levels we keep the original upper level and add a one parameter (the parameter is denoted by the symbol τ) family of new levels. These new levels involve more details than the original upper level. We call them UPPER levels. We construct them by taking a sharper view of the macroscopic system under investigation. We separate the particles composing it into two classes. Originally, the particles are indistinguishable (for instance all particles are white), now the particles are either white or red. The parameter τ∈R labels the extra degrees of freedom arising on UPPER levels due to sharper view of particles. Both passages *upper level* → *UPPER levels* and *UPPER levels* → *upper level* are thus known. The former is made by taking glasses allowing to recognize colors, the latter is made by becoming colourblind.

Besides the straightforward passage *UPPER levels* → *upper level* made simply by colour-blindness, there is another way to make the same passage. The extra degrees of freedom that arise on *UPPER levels* due to the sharper viewpoint are MaxEnt eliminated. In other words, we pass from *UPPER levels* to *upper level* in the same way as we passed from the upper level to the equilibrium level in the preceding section. The MaxEnt reduction of *UPPER levels* to the *upper level* will be termed *MaxEnt-reductions*, see [5].

A comparison of the upper reduced thermodynamic potential with the upper MaxEnt-reduced thermodynamic potential, both restricted to the critical region, is then the essence of the renormalization-group viewpoint of critical phenomena. Let the coefficients in the critical polynomials be ω for the upper reduced thermodynamic potential and Ω(τ) in the one parameter family of the upper MaxEnt-reduced thermodynamic potential. The difference between two UPPER levels, one corresponding to $\tau_{1}$ and the other to $\tau_{2}\neq \tau_{1}$ are manifested mathematically in $\Omega \left(\tau_{1}\right)\neq \Omega \left(\tau_{2}\right)$. The inseparability of UPPER levels in the critical point is mathematically expressed in Ω(τ) becoming independent of τ. Let $\Omega_{crit}$ be such a fixed point. Eigenvalues of the linearized renormalization dynamics (i.e., dynamics in which τ plays the role of the renormalization time—see the end of Section 2.3) are then the renormalized critical exponents.

The main features of this viewpoint of the renormalization-group theory of critical phenomena have already appeared in [8,27]. Also the illustration of the formulation presented below has been largely developed in [8]. In the original formulation of the renormalization-group theory [4] the upper level is the Microscopic level used as the upper level in the Gibbs equilibrium statistical mechanics. The state variable *x* in the Gibbs theory is the n-particle distribution function ($n∼10^{23}$ is the number of particles composing the macroscopic system under investigation). The upper reducing thermodynamic relation consists of the Gibbs entropy, the average microscopic energy, and normalization of the distribution function. The family of UPPER levels is constructed by extending the system in all directions by a scale factor τ. The MaxEnt-reduced levels are obtained by seeing the extension from the upper level to UPPER levels as replacement of every point with a “box” and the MaxEnt-passage from UPPER levels to the upper level as a MaxEnt reduction (with the Microscopic fundamental thermodynamic relation) of all boxes back to points.

The main difference between the original formulation of the renormalization group theory is thus the choice of the upper level. In the original formulation it is the Microscopic level. The Microscopic thermodynamic relation consists of the universal Gibbs entropy and an energy (Hamiltonian) in which the individual nature of the macroscopic system under investigation is expressed. The Ginzburg–Landau form of the energy is often used [28]. In our formulation the upper level is a general mesoscopic level and the upper reducing thermodynamic relation is its universal form (Landau polynomials arising the catastrophe theory) in the critical region. The main advantage of our formulation is thus its universal applicability and adaptability to dynamic critical phenomena.

### Illustration

In order to illustrate the renormalization-group theory of critical phenomena that is cast into the setting of the Landau theory we turn to the van der Waals theory recalled in Section 2.3. We keep the same equilibrium level and the same upper level. In addition we introduce an UPPER level with the state variables
(39)x=(f(r,v),g(r,v))
and the upper reducing thermodynamic relation
(40)$$\begin{array}{ccc} E^{↑}\left(f,g\right) & = & \int dr\int dv\left(\frac{v^{2}}{2}f\left(r,v\right)+\frac{v^{2}}{2}g\left(r,v\right)\right)\\ & & +\frac{1}{2}\int dr\int dv\int dr_{1}\int dv_{1}V_{pot}(|r-r_{1}\left|\right)\\ & & \times \left(f\left(r,v\right)f\left(r_{1},v_{1}\right)+g\left(r,v\right)g\left(r_{1},v_{1}\right)+2f\left(r,v\right)g\left(r_{1},v_{1}\right)\right)\\ N^{↑}\left(f,g\right) & = & \int dr\int dv(f(r,v)+g(r,v))\\ S^{↑}\left(f,g\right) & = & \int dr\int dv\left(-f(r,v)\ln f(r,v)-g(r,v)\ln g(r,v)\right.\\ & & \left.-f\left(r,v\right)\frac{\partial \theta }{\partial n(r)}-g\left(r,v\right)\frac{\partial \theta }{\partial m(r)}\right) \end{array}$$
where n(r)=∫dvf(r,v), m(r)=∫dvg(r,v), θ(n,m)=θ(n+m). The UPPER level represents a more detailed view of the van der Waals gas in the sense that the gas particles are no longer indistinguishable. They are divided into two groups. One group is composed of the same particles as on the upper level. We can call them now f-particles. Their states are characterized by the one particle distribution function f(r,v). The second group is composed of g-particles, the state variable is the one particle distribution function g(r,v). The f-particles and g-particles remain identical. In particular, interactions among the f-particles, among the g-particles, and among f-particles and g-particles are exactly the same as on the interactions of the f-particles on the upper level. The upper level differs from the UPPER level only in our ability to distinguish the f-particles from g-particles (for instance by having a different colour). The UPPER level thus indeed takes into account more details than the upper level. We are able to distinguish two colours.

We follow now the analysis that we made on the upper level in Section 2.3. We restrict ourselves to *n* and *m* that are independent of ***r*** and introduce the UPPER reducing thermodynamic potential
(41)$$\begin{array}{ccc} \Psi^{↑}\left(n,m;\beta ,A^{\left(n\right)},A^{\left(m\right)}\right) & = & n\ln n+m\ln m+\left(n+m\right)\theta^{\prime }\left(n+m\right)-\frac{1}{2}V_{pot}{\left(n+m\right)}^{2}\\ & & -\left(\ln A^{\left(n\right)}\right)n-\left(\ln A^{\left(m\right)}\right)m \end{array}$$
where we use the symbol Ψ instead of Φ to distinguish the UPPER level from the upper level and $\ln A=\alpha -\frac{3}{2}\ln\frac{\beta }{2\pi }$.

Next, we transform the UPPER level into a one parameter family of UPPER levels. We introduce first a one parameter family of the potentials $\Psi^{↑}$ by inserting into (Equation 41) $A^{\left(n\right)}=\;e^{-\tau }A;\;\;\;A^{\left(m\right)}=\;\left(1-\;e^{-\tau }\right)A$, where τ∈R;τ>0 is the parameter. The UPPER reducing thermodynamic potential (Equation 41) turns into the one parameter family
(42)$$\begin{array}{ccc} \Psi^{↑}\left(n,m;\beta ,A,\tau \right) & = & n\ln n+m\ln m+\left(n+m\right)\theta^{\prime }\left(n+m\right)-\frac{1}{2}V_{pot}{\left(n+m\right)}^{2}\\ & & -\ln\left(e^{-\tau }A\right)n-\ln\left(\left(1-e^{-\tau }\right)A\right)m \end{array}$$
where $\theta^{\prime }\left(n\right)=\frac{d\theta }{dn}$. We note that solution to $\Psi_{n}^{↑}=0$ is $e^{-\tau }n$, solution to $\Psi_{m}^{↑}=0$ is $(1-e^{-\tau })n$, and *n* is a solution to $\Phi_{n}^{↑}=0$ with $\Phi^{↑}$ given in (Equation 11). This means that if we are colour blind, then $n_{c}$ of the potentials (Equation 42) and (Equation 11) are the same. Also the reduced equilibrium thermodynamic relation implied by (Equation 42) and (Equation 11) are the same.

Now we pass from the UPPER level back to the upper level, see Figure 2. We can follow two routes:

Route 1:

On the first route we simply ignore the g-particles.
(43)$${\left[\Psi^{↑}\left(n,m;\beta ,A,\tau \right)\right]}_{m=0}=\Phi^{↑}\left(n,A,\tau \right)$$
where $\Phi^{↑}$ is given in (Equation 11)

Route 2:

On the second route we eliminate the presence of g-particles with MaxEnt. In this way we arrive at
(44)$${\left[\Psi^{↑}\left(n,m;\beta ,A,\tau \right)\right]}_{\Psi_{m}^{↑}=0}$$

We have transformed f-particles into f-quasi-particles, i.e., f-particles that are modified by taking into account the presence of g-particles. The passage from particles to quasi-particles is a pattern recognition process. The more are the quasi-particles different from the original particles the more pronounced is the pattern.

Following our terminology and notation (we recall that the thermodynamic potential on the upper level is denoted by the symbol Φ) we denote the potential (Equation 43) by the symbol $\Phi^{↓}$ and call it upper reduced potential. The potential (Equation 44) is denoted $\Phi^{\left(↓ME\right)}$ and called upper MaxEnt-reduced potential in order to point out its provenance.

We expect that the two upper thermodynamic potentials (Equation 43) and (Equation 44) are different. The former is just the original upper reducing potentials (Equation 42) without the g-particle. The latter is also the upper reducing potential (Equation 42) without the g-particles but with their presence felt through MaxEnt. The passage from the UPPER potentials (Equation 42) to the upper MaxEnt-reduced potential (Equation 44) can be seen as a process of recognizing a pattern on the UPPER level. The recognized pattern is then expressed in terms of the upper-level state variables. In the absence of such patterns on the UPPER level the upper MaxEnt-reduced potential (Equation 44) will be the same as the upper reduced potential (Equation 43). This is expected to happen when the inclusion of more details on the UPPER level does not reveal anything new (new with respect to what is seen on the upper level). The experimentally observed inseparability of levels in the critical region then suggests to define the critical point as the point at which the potentials (Equation 43) and (Equation 44) are identical.

In order to be able to compare the critical part of the upper MaxEnt-reduced potential (Equation 44) with the critical part of the upper reduced potential (Equation 43), we cast them into the form of the Landau polynomials. If we choose (Equation 23) for the renormalization time evolution of $\Psi_{crit}^{\left(↓ME\right)}$, then the upper MaxEnt-reduced thermodynamic potential takes the form
$$\Phi_{crit}^{\left(↓ME\right)}\left(n,\beta ;A,B,\tau \right)=e^{-\chi \tau }\left(\Omega_{1}\left(\Omega ,\chi ,\tau \right)\zeta +\Omega_{2}\left(\Omega ,\chi ,\tau \right)\zeta^{2}+\Omega_{3}\left(\Omega ,\chi ,\tau \right)\zeta^{4}\right)$$
where $\Omega =(\Omega_{1},\Omega_{2},\Omega_{3})$ is a solution of (compare with (Equation 24))
(45)$$\dot{\Omega }=R^{\left(ME\right)}\left(\Omega ,\chi \right).$$

It remains to show the relation of Ω,ζ and $R^{\left(ME\right)}$ to (n,β,A,B) appearing in the upper potential (Equation 43) and to investigate the renormalization time evolution governed by (Equation 45).

Regarding the former task, we only indicate the route and refer to [8] for details. We recall that solution to $\Psi_{n}^{↑}=0=\Psi_{m}^{↑}$ is $n=e^{-\tau }n^{\Phi^{↑}}$, $m=\left(1-e^{-\tau }\right)n^{\Phi^{↑}}$, where $n^{\Phi^{↑}}$ is a solution to $\Phi_{n}^{↑}=0$ with $\Phi^{↑}$ given in (Equation 11). This means that the reduced equilibrium thermodynamic relation implied by (Equation 42) and (Equation 11) are the same.

Next, we note that
(46)$$\Phi^{\left(↓ME\right)}\left(n;\beta ,A\right)=\Phi^{↑}\left(n;\beta ,e^{-\tau }A\right)-\tau ne^{-\Phi_{n}^{↑}\left(n;\beta ,A\right)}+O\left(\tau^{2}\right)$$

This relation follows from
$$\Psi_{m}^{↑}=0\Rightarrow \ln\left(\frac{m}{n}\right)=-\Phi_{n}^{↑}\left(n;\beta ,A\right)+\ln\left(\frac{A^{\left(m\right)}}{A}\right)$$
and from $\Psi^{↑}\left(n,m;\beta ,A^{\left(n\right)},A^{\left(m\right)}\right)=\Phi^{↑}\left(n;\beta ,A^{\left(n\right)}\right)-m+O\left(\tau^{2}\right)$. The remaining details can be found in [8].

Regarding the latter task, solutions to the renormalization time evolution governed by (Equation 45) have been investigated in [8]. Three fixed points have been identified and the largest eigenvalue of the linearized (Equation 45) about one of them equals 0.8. The largest eigenvalue in (Equation 25) (which is the classical critical exponent corresponding to the approach to the coexistence line in a transverse direction) is 0.75. The value of this type of critical exponent measured in experiments is indeed close to 0.8 [8,29]. The physical significance of this agreement remains to be investigated. Our main objective in this section was to illustrate the renormalization-group theory of critical phenomena in the setting of the Landau theory of critical phenomena.

## 5. Concluding Remarks

Thermodynamics is a theory of relations among theories of macroscopic systems formulated on different autonomous levels of description. Our main objective in this paper is to show that this multiscale viewpoint of thermodynamics unifies investigations of static and dynamic critical phenomena. We emphasize that the reduction of an upper level to a lower level (a level involving fewer details than the upper level) represents a loss of details but a gain of emerging features arising as patterns in the phase portrait of the upper level. Applicability of multi-level thermodynamics is ubiquitous, ranging from chemical engineering, rheology, electrodynamics of matter, kinetic theory to machine learning [30].

Classical equilibrium thermodynamics arises in investigations of relations between an upper level (e.g., the microscopic level or the level of kinetic theory) and the equilibrium level, on which no time evolution takes place. The upper reducing entropy, which generates the approach to the equilibrium level, becomes the equilibrium entropy when the approach is completed. When the approached lower level involves the time evolution, the result of the reduction can either be seen as a reduction in the upper state space (approach to a quasi-invariant submanifold representing the lower state space in the upper state space) or as an approach of the upper vector field to the lower vector field). The thermodynamics that arises by following the latter viewpoint is termed rate-thermodynamics. The mathematical formulations of thermodynamics and rate-thermodynamics are essentially identical. The thermodynamic potentials are however replaced with rate-thermodynamic potentials. In the particular case of externally unforced systems, when the lower level is allowed to reach the equilibrium level, the upper rate-entropy is closely related to the production of the entropy generating the approach of the lower level to the equilibrium level.

The multiscale thermodynamics acquires two new features in the critical region. First it is the universality of the thermodynamic and rate-thermodynamic potentials and the inseparability of levels. The former is a consequence of the mathematical representation of criticality (catastrophe theory) and the latter the consequence of the physical nature of the criticality. The former feature has been noted by Lev Davidovich Landau and is a basis of his theory of critical phenomena. The latter feature is a basis of the renormalization-group theory of critical phenomena. We show that the multiscale thermodynamics provides a unified setting for the Landau theory of static critical phenomena, for its extension to the dynamic critical phenomena and for the renormalization-group theory of critical phenomena.

## Figures and Tables

**Figure 1 entropy-22-00978-f001:**
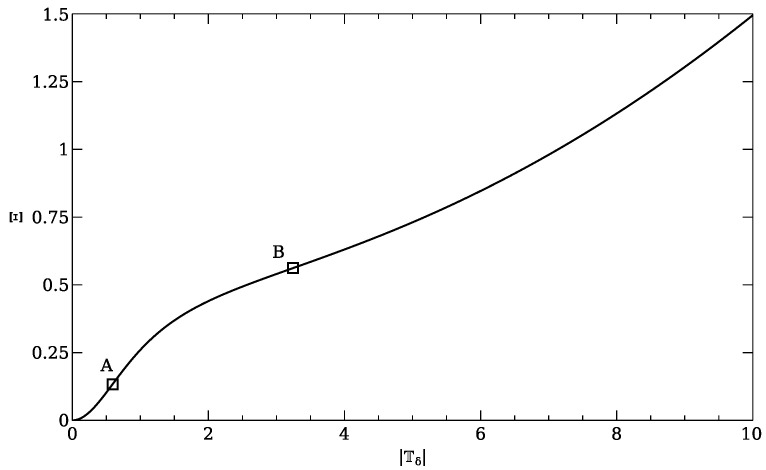
Dissipation potential (Equation 31). The points where convexity of lost are highlighted. This potential was found in [24] and is based on a work by Le Roux and Rajagopal [25]. The rate thermodynamic analysis of the qualitative implied by the potential was then confirmed by numerical simulations in [26].

**Figure 2 entropy-22-00978-f002:**
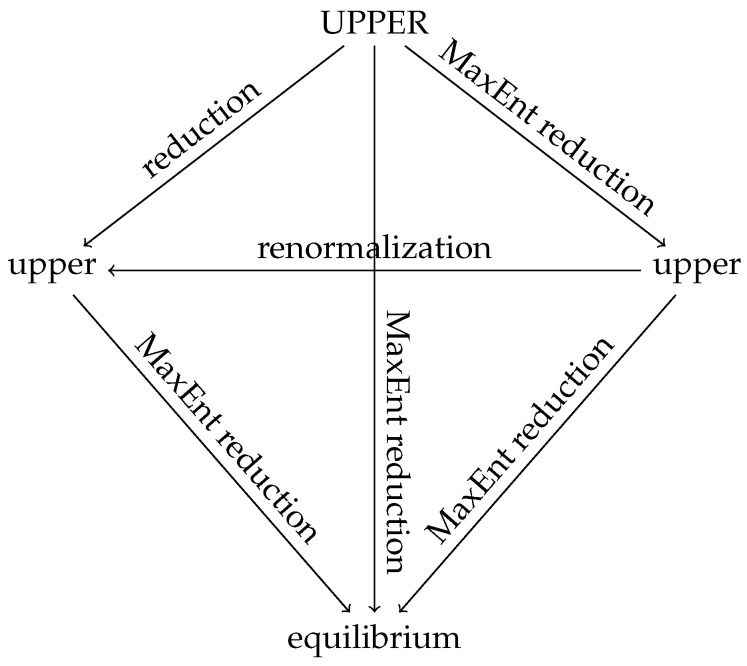
Diagram of the levels of description.

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
