# Peer review of "Dynamic and Renormalization-Group Extensions of the Landau Theory of Critical Phenomena"

_entropy, 2020, doi:10.3390/e22090978_

Round 1

Reviewer 1 Report

I recommend accepting the manuscript after the authors do the major revision included in the review comments in the attachment file.

Author Response

Please find the attahed pdf.

Reviewer 2 Report

The paper is devoted to the theoretical description of critical phenomena at phase transformations. The mathematical development is based on the well-known Landau theory of phase transition. The equations set, characterizing the different approaches of the description of phase transition, are given. In my opinion, this manuscript can be published in Entropy after revision.

Questions and remarks.

  1. It is necessary to indicate the limitations of the present theory, in particular, in an impossible description of the crystal/liquid transition, where the critical point does not exist.

  1. It is necessary to indicate how the experimental study can test the present theory.

  1. Page 2, line 41. What does it mean that fluctuations are invisible on the equilibrium level? It should be clarified.

  1. Page 3, lines 113-114. The examples of suitable quantities as x-parameter should be given.

  1. Page 5, Eq. 10. The variable n in Equation 10 should be defined.

  1. Page 6, line 182. It should be clarified which variable – the temperature or the beta (an inverse temperature) – is considered to be the lowest.

  1. Page 6, Eq. 15. The variable nc in Equation 10 should be defined.

  1. Page 6, line 192. The sentence “This for of fundamental thermodynamic relation” should be reformulated.

  1. Page 7, line 203. The definition of “renormalization time evolution” should be given.

  1. Page 9, lines 278-280. “For instance the experimentally observed behavior of Rayleigh-Bénard system (a thin horizontal layer of a fluid heated from below)is well described on the level of hydrodynamics (in Boussinesq equations).”

The reference should be given.

  1. Page 13, lines 399-400. “The Ginzburg-Landau form of the energy is often used.”

The reference should be given.

  1. Page 13, lines 404. “In addition we introduce in addition”

The phrase should be modified.

  1. Page 14, Eq. 39. It is necessary to define the quantitate n in Equation 39.

  1. Page 15, line 434. The definition of “quasi-particles” should be given.

  1. Page 16, line 463. “The value of this type of critical exponent measured in experiments is indeed close to 0.8.”

The reference should be given.

  1. Page 17, line 504. The typo in Ref 1: pase -> phase.

Author Response

Pleased find the attached pdf.

Round 2

Reviewer 1 Report

The authors have satisfactorily responded to all the questions and made the necessary changes to the manuscript. I have no further questions and suggest the acceptance of the revised manuscript.